# TERRA and Telomere Maintenance in the Yeast *Saccharomyces cerevisiae*

**DOI:** 10.3390/genes14030618

**Published:** 2023-02-28

**Authors:** Bechara Zeinoun, Maria Teresa Teixeira, Aurélia Barascu

**Affiliations:** Laboratoire de Biologie Moléculaire et Cellulaire des Eucaryotes, CNRS, UMR8226, Institut de Biologie Physico-Chimique, Sorbonne Université, F-75005 Paris, France

**Keywords:** TERRA, R-loops, senescence, telomere maintenance, survivors, *Saccharomyces cerevisiae*

## Abstract

Telomeres are structures made of DNA, proteins and RNA found at the ends of eukaryotic linear chromosomes. These dynamic nucleoprotein structures protect chromosomal tips from end-to-end fusions, degradation, activation of damage checkpoints and erroneous DNA repair events. Telomeres were thought to be transcriptionally silent regions because of their constitutive heterochromatin signature until telomeric long non-coding RNAs (LncRNAs) were discovered. One of them, TERRA (TElomeric Repeat-containing RNA), starts in the subtelomeric regions towards the chromosome ends from different telomeres and has been extensively studied in many evolutionarily distant eukaryotes. Changes in TERRA’s expression can lead to telomeric dysfunction, interfere with the replicative machinery and impact telomere length. TERRA also co-localizes in vivo with telomerase, and can form RNA:DNA hybrid structures called R-loops, which have been implicated in the onset of senescence and the alternative lengthening of telomere (ALT) pathway. Yet, the molecular mechanisms involving TERRA, as well as its function, remain elusive. Here, we review the current knowledge of TERRA transcription, structure, expression, regulation and its multiple telomeric and extra-telomeric functions in the budding yeast *Saccharomyces cerevisiae*.

## 1. Introduction

Chromosomes are subjected daily to constant genotoxic stresses, which can be of endogenous (replicative stress, oxidative stress, etc.) or of exogenous origin (irradiation, exposure to chemical agents, etc.). Among the different types of DNA damage, double stranded breaks are the most toxic because they can lead to the loss of large pieces of chromosomes. DNA damage repair mechanisms ensure genome stability through the orchestration of a set of enzymatic activities, which specifically identify these accidental double-stranded ends and repair them, thus limiting the loss or alteration of genetic information for future generations. However, since eukaryotic chromosomes are linear, they have natural ends that resemble double-strand breaks called telomeres. The main function of these nucleoprotein structures is to maintain the integrity of chromosomes by protecting them from the DNA repair machinery. Thus, functional telomeres are not recognized as accidental double-strand breaks and do not become fused or degraded by repair processes [1]. 

The nucleotide sequence of telomeres is highly conserved in eukaryotes, formed by regular or irregular tandem repeats of a guanine-rich motif, which terminates in most organisms studied with a 3′ tail. This composition is due to the activity of telomerase, the main enzyme ensuring telomere length maintenance. In budding yeast, telomeres are approximately 300 +/− 75 b and telomeric DNA is histone-free, formed by an irregular repeated sequence T(G)_1−3/_(C)_1−2_A and terminates with a 5 to 10 nt 3′ single-strand overhang [2,3,4]. The length of the telomeric single-strand region is also dynamic and heterogeneous in nature, increasing transiently in the late S phase (about 40–100 nt) [2,5,6,7]. The protection and stability of the telomere structure is ensured by a specialized protein complex referred to as “shelterin” in mammalian cells. In *S. cerevisiae*, Rap1 was first described as a transcription regulator of many essential genes across the yeast genome, but also binds double-stranded telomeric DNA to protect chromosome ends from fusions. At telomeres, Rap1 recruits Rif1/2 proteins, which contribute to telomere protection functions and to regulating telomerase. In addition, Rap1 recruits the Sir3/4 (silent information regulator) proteins, which interact with Sir2, a NAD+-dependent histone deacetylase, to repress transcription. The CST complex, found at the 3′ overhang, comprises Cdc13, Stn1 and Ten1 to protect the C-strand from 5′ to 3′ resection, and regulates telomerase [8].

To better understand the essential functions of telomeres, we must decipher the dynamics and changes in telomere length, protein composition and structure along the cell cycle and over consecutive cell divisions. In fact, DNA replication is one of the mechanisms that inevitably affects telomeres. The enzymatic features of DNA polymerases do not support the complete replication of telomeres, generating the so-called end replication problem. This causes a progressive shortening of telomeres with each cell cycle. Telomerase is a reverse transcriptase that serves to counteract this end-replication problem in most eukaryotes studied to date. In budding yeast, telomerase is composed of a catalytic reverse transcriptase subunit Est2, two accessory subunits, Est1 and Est3, and a long non-coding RNA, TLC1, that serves as a template for the iterative synthesis of telomeric repeats. Telomerase is constitutively expressed in yeast cells to maintain telomeres at an average size that results from a balance between permanent shortening and lengthening and to enable unlimited yeast proliferation [9]. However, in many multicellular organisms such as humans, only few cells including stem and germ cells express telomerase. Somatic cells repress telomerase, so telomeres shorten with each cell division until they reach a critical size that leads to a permanent arrest of proliferation called replicative senescence. At the molecular level, this arrest is caused by the activation of the DNA damage response at telomeres. In *S. cerevisiae*, the senescence signal in telomerase-negative cells is thought to come from the first telomere reaching a critical reduced size [10,11], an idea which was consolidated by mathematical simulations and formal genetics [12]. Consistent with this hypothesis of senescence being driven by the shortest telomeres, five dysfunctional telomeres were detected in human cells in culture at the onset of senescence [13]. Thus, telomeres play a central role in controlling cell viability, and their proper functioning ensures genome stability.

Subtelomeres are genomic regions found upstream of the telomeres and are made up of long, heterogeneous and variable repeated elements, subject to rapid evolution. Subtelomeric regions do not contain genes essential for cell viability under rich growth conditions, but the extreme plasticity of these regions ensures rapid adaptation of yeast under various stress conditions [14,15,16,17]. This is because subtelomeres include multigene families coding for functions related to interaction with the environment. Unlike telomeres, the subtelomeric regions of *S. cerevisiae* are covered by nucleosomes [18,19]. The Sir proteins play a key role in the compaction of these regions and the telomere position effect (TPE) that results in the silencing of genes near telomeres [20,21]. In *S. cerevisiae*, all subtelomeric regions have an X element, followed by one to four repeats of a 5–7 kb-long Y′ element in nearly half of the telomeres [22,23,24,25]. However, despite the heterochromatin marks found in these regions, the subtelomeric sequences and even telomeres are transcribed into different non-coding RNAs [26,27,28]. Telomeric transcription was described in *S. cerevisiae*, *Schizosaccharomyces pombe* and several other mammals, birds, fish, plants and protozoa species, suggesting that telomeric transcription spans the whole eukaryote evolution [27,29,30,31,32,33,34,35,36,37,38,39]. Among the non-coding RNAs transcribed from the regions closer to the tip are ARRET, αARRET, subTERRA and TERRA. Here, we focus on the most recent findings on TERRA in *S. cerevisiae*.

## 2. TERRA Transcription and Primary Structure

TERRA (TElomeric Repeat-containing RNA) was first discovered in mammalian cells as a long non-coding RNA which contains UUUAGG repeats and varies in length from 100 b to more than 9 kb, colocalizes with telomeres, and is considered part of telomeric heterochromatin [26,40]. TERRA transcription at telomeres is an apparently conserved feature in eukaryotes, and its length and quantities vary within different cell lines, species, tissues and developmental stages. In budding yeast, Northern blot analysis showed that TERRA is heterogeneous, ranging from 100 to 1200 b, with an average of 380 b. In *S. cerevisiae*, TERRA levels are kept low mainly by the 5’-3’ exonuclease Rat1, a fraction of which is found at telomeres and degrades TERRA [36,41]. Xrn1, the cytoplasmic paralog of Rat1, has no effects on TERRA levels, supporting the idea that yeast TERRA is mostly nuclear.

TERRA LncRNA is transcribed by RNA Polymerase II, using the telomeric C-rich strand as a template (Figure 1). ChIP experiments revealed that RNA pol II associates with telomeres, and inactivating this polymerase by an *rpb3-2* temperature-sensitive allele blocks TERRA accumulation [36]. TERRA transcription starts at subtelomeres, toward the end of the chromosomes from both Y′ and X-only telomeres. Thus, TERRA has the same sequence as the G-rich sequence and harbors specific subtelomeric sequences at the 5′ end and telomeric repeats at the 3′ end [36,41]. Most of TERRA heterogeneity stems from its 3′ ends, yet its promoter remains ill-defined. Using 5′RACE and RLM-5′RACE, some transcription start sites (TSSs) of TERRA have been identified on several telomeric ends from Y′ and X-only telomeres [42,43,44]. This revealed that Y′ TERRA starts closer to telomeric repeats than X-only TERRA and that the +1 nucleosome for TERRA transcription is stabilized by Sir2. In mammalian cells, promotors have been found at numerous human chromosome ends in CpG-islands residing in subtelomeric regions, and TERRA was similarly shown to be affected by the chromatin state [45,46]. TERRA accumulates at the G1/S transition prior to the replication of telomeres and then decreases in the late S phase. Thus, in budding yeast, similarly to mammalians cells, TERRA is finely regulated through the cell cycle [47,48,49].

The canonical poly(A) polymerase Pap1 polyadenylates the majority of yeast TERRA, bestowing more stability to the molecule [36]. In a *pap1-1* temperature-sensitive allele, TERRA is destabilized at a restrictive temperature. CF1A, the endo-nucleolytic cleavage factor that acts prior to Pap1, seems to also be important for TERRA biogenesis [36] (Figure 1). In contrast, the deletion of *TRF4*, encoding an alternative poly(A) polymerase of the TRAMP complex (whose action targets RNAs for degradation by the exosome) increases TERRA levels in a background where TERRA already accumulates, indicating that TERRA can be polyadenylated by TRAMP and thus subjected to nuclear exosome degradation, but to a minor extent [36,50] (Figure 1). Consequently, TERRA would harbor a poly(A) tail on its 3′ end. Yet, only 7% of yeast TERRA is polyadenylated [50]. In human cells TERRA, the absence or presence of the poly(A) tail determines the association or not to chromatin [26,36,38,40,41,49]. In humans and fission yeast, TERRA harbors at its 5′ end a 7-methylguanosine cap structure not yet identified in budding yeast [49,51].

The low levels detected in wild-type yeast cells makes it difficult to quantify endogenous TERRA by Northern blot and RT-qPCR [36]. Thus, to further study TERRA biogenesis and functions, TERRA levels have been artificially raised using artificial regulatable promotors inserted before the telomeric tract or in subtelomeric regions [36,42,52,53]; but, these methods have shown that it interferes with Ku70/80 function and creates an Exo1-dependent resection at the chromosome ends (see below). Another approach was through in vivo detection of endogenous TERRA molecules. Several 21 nt-long stem-loop MS2 phage sequences were inserted in tandem within the TERRA locus of a single telomere. Upon transcription, the resulting TERRA molecules can then be detected by live cell imaging through the high-affinity binding of the MS2 phage coat protein fused to a fluorescent protein expressed in cells. Using this approach, it was concluded that TERRA is detected only in a small population of cells (10%) from each given telomere, and the transcription product forms a perinuclear focus during the interphase [54]. In other words, for each given chromosome end, TERRA appears to be stabilized at telomeres approximately one time every ten cell divisions. Thus, in each cell, ~10% of telomeres would have their TERRA stabilized.

## 3. Transcriptional, Co-Transcriptional and Post-Transcriptional Regulations

Subtelomeric elements control transcription in many ways, and this was first discovered through the study of the silencing of polymerase II-dependent genes inserted near telomeres: the so-called “telomere position effect” (TPE). TERRA transcription itself is subjected to some known silencing mechanisms, such as the ones regulated by the Sir complex. Another way subtelomeres limit TERRA transcripts’ level is by regulating distinct Rap1-mediated pathways (Figure 1) [41]. This was discovered by detecting an increase in TERRA levels at both X-only and Y′ telomeres in a *rap1-17* mutant in which the C-terminus of Rap1 was truncated, thus preventing the recruitment of both Rif1/2 and Sir2/3/4 complexes to telomeres. When dissecting which protein complex was responsible for this effect, it was found that at the X-only telomeres, Rap1 recruits both the Sir2/3/4 complex, which presumably represses TERRA transcription, and the Rif1/2 complex, which negatively regulates TERRA in part via Rat1-dependent degradation. In contrast, at the Y′ telomeres, the Sir2/3/4 complex has no effect on TERRA levels and the Rif1 effect is independent of Rat1. Subtelomeres also contain binding sequences for Reb1, an essential transcription factor. In *reb1^ts^*, TERRA levels increase 75-fold compared with WT, suggesting an effect of Reb1 in the transcription of TERRA (Figure 1) [43]. Since most subtelomeric binding sites of Reb1 reside downstream of the TSS of TERRA in the X element, it appears that Reb1 represses TERRA levels by a roadblock transcriptional termination mechanism, independently of Sir-complex-dependent chromatin modifications. The PAF1 complex is a transcription elongation factor that affects RNA levels through transcriptional and post-transcriptional modifications and via its binding to RNApol II. In budding yeast, this complex is recruited at telomeres and limits TERRA levels through its subunits Paf1 and Ctr9 [50]. While the exact mechanism, possibly co-transcriptional, is still elusive, it appears that it is mostly independent from Rat1, the Sir complex and the exosome (Figure 1). The SMC5/6 complex, one of the structural maintenance of chromosome protein complexes and which plays an organizational and structural role in telomeres, is also involved in limiting TERRA levels [55]. While this complex is involved in the recruitment and sumoylation of the Sir complex, the effects of SMC5/6 complex on TERRA levels are also partially independent from Sir4 (Figure 1).

Overall, at all telomeres, the Sir complex and Rat1 are the key to limiting the global levels of TERRA, but many other factors are involved in fine tuning TERRA quantities in cells. Still, the additional level of regulation of TERRA within the S phase was found to depend mainly on Rat1 [47].

## 4. TERRA RNA:DNA Hybrid Regulation

TERRA was found to form three-stranded RNA:DNA telomeric hybrid structures, called R-loops, whose regulation is linked to telomere size and homeostasis [36,47,56,57,58]. R-loops are RNA:DNA structures in a three-stranded configuration where the RNA hybridizes with the DNA template, leaving a single-stranded DNA loop. TERRA telomeric R-loops are preferentially formed due to the G-rich nature of TERRA that binds stably to the C-rich DNA template. In addition, R-loops are naturally stable since the interaction of RNA:DNA is conformationally stronger than DNA:DNA. Depending on the levels and the location of R-loops, they could be beneficial or deleterious to the genome [59]. R-loops can be a source of replication stress, interfering with the DNA replication machinery and leading to replication fork collapses and collisions. This was shown to generate DSBs triggering recombination events [60]. In contrast to TERRA, which is hard to detect in unchallenged cells, RNA:DNA hybrids are readily detected at telomeres by chromatin immunoprecipitation using an antibody (S9.6) that recognizes the RNA:DNA hybrids, known as the DRIP technique (DNA-RNA ImmunoPrecipitation) [47,56,57,58]. Using this technique, it was found that RNA:DNA hybrids accumulate in the early S-phase, before telomere replication, which is known to happen later in the S-phase, suggesting that TERRA and R-loops are regulated throughout the cell cycle in a coordinated way with replication such that the potential harmful outcomes of the TERRA R-loops are avoided [47]. Recently, R-ChIP experiments using a catalytic dead form of RNase H1, Rnh1-cd, yields more sensitivity in detecting hybrids at telomeres and could be used to detect more layers of TERRA regulation [61].

RNase H1 and RNase H2 enzymes contribute to the removal of RNA:DNA hybrids by endonucleolytic cleavage of the hybridized RNA [62,63]. While RNAse H1 is composed of the gene product of *RNH1*, the RNase H2 holoenzyme is made of three subunits: Rnh201 (the catalytic one), Rnh202 and Rnh203. Additionally, while RNAse H1 functions independently of the cell cycle and mostly responds to R-loop stress, RnaseH2 may only be effective in certain cell cycle phases. The deletion of RNase H enzymes and the OE of RNase H1 are mostly used to control R-loop levels in the cell. The first evidence that TERRA forms telomeric R-loops in budding yeast comes from the overexpression of Rnh201, which was found to specifically reduce TERRA in a *rat1-1* mutant in which TERRA accumulates [36]. Conversely, in an *rnh1 rnh201* double mutant, R-loop levels increase [53,56,57], suggesting that telomeric R-loops are controlled by endogenous RNase H enzymes (Figure 1). Nevertheless, most of TERRA in a wild-type cell appears not to be in an R-loop configuration, since the over-expression of RNase H1 only reduces TERRA at some telomeres. Most importantly, RNase H2 participates in the regulation of TERRA R-loops during the S phase [47]. The telomeric protein Rif2 contributes to RNase H2 recruitment to degrade telomeric R-loops, and accordingly the deletion of Rif2 causes the accumulation of hybrids, showing its importance in preventing R-loop formation. In contrast, RNase H1 could not be detected at telomeres in telomerase-positive cells, suggesting that TERRA R-loops are mainly removed by RNase H2 in unchallenged conditions.

The THO complex plays a role in transcription elongation and mRNA export. In budding yeast, four proteins constitute this complex: Tho2, Hrp1, Mtf1 and Thp2 [64]. A fraction of this complex was found at the telomeres and plays a role in telomere stability [57]. Indeed, THO promotes the telomere maintenance through the control of TERRA biogenesis. Similar to an *rnh1 rnh201* double mutant, a *thp2* mutant has no effect on the global pool of TERRA, but shows an increase in telomeric RNA:DNA hybrids. In contrast, the deletion of *THO2* and *HRP1* downregulates the levels of TERRA at both Y′ and X-only element telomeres and increases TERRA hybrids at telomeres [53,57,65]. THO complex deregulation can thus lead to an inappropriate assembly of TERRA into native ribonucleotides, thereby impairing its localization. Therefore, this complex plays a role in TERRA processing and suppresses R-loops at telomeric levels (Figure 1).

The SMC5/6 complex not only limits the level of TERRA (as discussed above) but can also act on R-loops through the negative regulation of Mph1, the helicase yeast homolog of mammalian FANCM. Mph1 was found to accumulate at RNA:DNA hybrids, which increase in RNAse H mutants during replication to prevent the accumulation of DNA damage [58]. In addition, Mph1 accumulates at short telomeres in an R-loop-dependent manner. The model would show that Mph1 is recruited when R-loops accumulate to participate in recombinogenic activities triggered by R-loops. The SMC5/6 complex would thus be essential for the regulation of Mph1 to limit its pro-recombinogenic activity at RNA:DNA hybrids.

Other factors may also be implicated in the fine regulation of TERRA telomeric R-loops including the Sen1 and Pif1 RNA/DNA helicases that may locally unwind these hybrids. The nonsense-mediated mRNA decay pathway (NMD) involved in the RNA quality control surveillance mechanism may also regulate TERRA. NMD mutants have shorter telomeres because of an increased level of telomere-capping factors Stn1 and Ten1 [66,67]. High-throughput analysis shows that Y′ element transcripts accumulate in an NMD mutant (*upf1Δ*) [68], but Northern blot analysis reveals that TERRA transcripts do not increase in NMD mutants (*upf1Δ*, *upf2Δ*, *upf3Δ*, *xrn1Δ*) [36]. This does not exclude that the NMD pathway could regulate TERRA association to telomeres locally without impacting the total amounts, as observed in mammalian cells [26].

## 5. TERRA and Telomeres Length

It has become clear that the TERRA pool and R-loop levels were found to increase when telomeres are shorter and increase even more when telomeres become critically short. This *cis* regulation, dependent on the amounts of Rap1 and Rif2 in cis, is dependent on telomere length, as detailed in the above sections [41,47,69]. Rat1 and RNase H2 are involved in this specific regulation, and the loss of telomeric proteins with the shortening of telomeres impairs their recruitment, resulting in more TERRA [47,69]. Thus, TERRA is upregulated at short telomeres.

Yet, in a telomerase-positive context, telomere length and TERRA levels were found to not always correlate inversely in mutants affected by the homeostasis of telomere length. *rat1-1, paf1Δ, nse3-1 and thp2Δ* mutants have shorter telomeres and increase the TERRA pool or accumulate R-loops [36,50,57]. On the contrary, some mutants show a positive correlation between TERRA levels and telomere length and others are neutral. For example, *rap1-17* and *Rif1/2Δ* mutants have longer telomeres and increase the TERRA pool, but Sir2/3 have milder changes in telomere length [41]. Mutants in the THO complexes *tho2Δ* and *hpr1Δ* have longer telomeres and show an increase in R-loops [57]. Over-elongated telomeres by tethering telomerase to telomeres only show a slight decrease in TERRA levels [41]. If we take a specific case, *sir2Δ* and *reb1^ts^* have longer telomeres and more TERRA, but the double mutants have even more TERRA and shorter telomeres compared with single mutants. Therefore, one cannot rule out that some proteins involved in TERRA regulation can have other functions, which could impact telomeres independently of regulating TERRA.

## 6. TERRA Interacts with Telomerase and Regulates Its Function

TERRA expression increases with the shortening of telomeres in a cis regulation (as discussed above) and telomerase is recruited preferentially to short telomeres [70]. This is due to the increased occupancy of Rap1 molecules at longer telomeres inhibiting telomerase [71]. Since TERRA is transcribed from the C-rich strand and it contains telomeric repeats, one can speculate an interaction between TERRA and the telomerase template RNA, the *TLC1* LncRNA, which is complementary to TERRA. In budding yeast, telomerase impaired action was first suspected to be the cause of short telomeres observed in a *rat1-1* mutant where TERRA levels increase [36]. Indeed, telomere length in the *rat1-1 est2Δ* double mutant is epistatic to an *est2Δ* single mutant, indicating that telomerase-dependent lengthening of telomeres is impaired in *rat1-1* cells. The overexpression of RNaseH2 in the *rat1-1* context leads to less TERRA and suppresses the phenotype observed, which could suggest that TERRA:DNA hybrids may inhibit telomerase. However, TERRA telomeric hybrids were not detected specifically; rather, TERRA global levels were measured by Northern blot, and since RNase H2 works in a trimeric complex, we do not know if its over-expression actually reduces R-loops and, if so, to what extent. Further studies are needed to bring a clear conclusion on this matter.

Using live-cell imaging, the laboratory of P. Chartrand showed that TERRA expressed from a given telomere forms a perinuclear focus, which becomes multiple foci when telomeres shorten in the absence of telomerase [54,72]. TERRA foci colocalize and interact with *TLC1* RNA independently of DNA during the G1/early S phase preceding the clustering of telomerase during the late S phase, preferentially at the same telomere that transcribed TERRA. This suggests that TERRA would be the signal generated by these short telomeres, playing a role in the nucleation of *TLC1* RNA and the clustering of telomerase required for telomere elongation in cis [73]. Thus, TERRA could be assigned the role of a scaffold molecule. Evidence that TERRA interacts physically with telomerase and promotes its recruitment and activity in telomeres has been also reported in fission yeast [74].

In mammalian cells, TERRA interacts with telomerase and its template LncRNA in vivo [75]. Short TERRA-like oligonucleotides can base pair with telomerase template RNA, interact with telomerase and reduce the activity, but not the processivity, of telomerase in vitro [40,75,76]. In contrast, in vivo studies do not support that notion, showing that telomerase elongates telomeres efficiently independently of TERRA and telomere transcription [46]. This does not exclude the fact that, in vivo, factors may exist participating in the switching of telomerase from TERRA to telomeric ends, repressing its inhibition depending on the cell-cycle phase [76,77].

All these studies agree that TERRA interacts with telomerase, but further investigation is required to draw a clear conclusion on how this interaction plays a role in telomere length maintenance and to unveil the detailed mechanisms behind it (Figure 2).

## 7. TERRA and Telomere Replication

Telomeres are replicated through semiconservative DNA replication and, if present, telomerase contributes to the maintenance of their length. Replication is tricky and when it comes to telomeres, considering their heterochromatin and repetitive structure, it is even more complicated, as slippage events may occur. Special helicases and telomere binding proteins are required for a complete replication [78,79,80,81,82]. In case replication forks stall and collapse, rapid loss of entire telomeric tracts can happen. Yet, telomeres display an intrinsic propensity for controlled degradations at the time of replication. Upon the passage of the replication fork, a physiologically limited 5′-3′ resection occurs at the tip of the leading strand to reform the 3′-G-overhang, which is essential for telomere functions at every passage of the replication fork [2,6]. The MRX-Sae2 complex plays a role in this end processing, followed by the action of Sgs1-Dna2 and Exo1 [83,84]. However, Exo1 can also contribute to unscheduled telomere end resection when telomeres lose their protection function in a mutant of the telomeric proteins Cdc13 or Ku70/80 [85,86,87,88,89]. Thus, TERRA and TERRA R-loops need to be finely regulated to avoid being a source of telomeric double-strand breaks or DNA degradations and loss of telomeric tracts. The cell cycle regulation of TERRA and TERRA R-loops, which accumulate early in the S-phase and decrease afterwards, necessarily contribute to preventing such deleterious events at the time of the passage of the replication fork [47]. Yet, many factors contribute to keeping TERRA in check to prevent telomere replication-related problems.

The notion that TERRA interferes with replication came from two sets of experiments in which TERRA was artificially induced. The first one used a galactose inducible promotor, upstream activation sequence (UAS), just in front of telomeric repeats on telomere 7L (7L-Gal tiTEL), resulting in telomeric transcripts harboring only telomeric repeats [90]. The second one used a doxycycline regulatable promotor, upstream TERRA transcription start site on chromosome 1L (TetO7-1L tiTEL), thus resulting in the expression of full-length TERRA [42]. The result in both cases was a transcription-induced telomere shortening in cis, specific to the telomere from which the transcription is enhanced. Furthermore, R-loop levels did not change upon the forced induction of TERRA. These elements suggest that TERRA per se does not have a trans effect at other telomeres, but rather high levels of transcription within telomeric repeats could cause the shortening. The shortening observed was found to be independent of telomerase inhibition or the binding of telomeric proteins to telomeres (Ku70/80 and Rap1) [42]. In contrast, the shortening was found to be dependent on the S phase of the cell cycle, and thus possibly linked to the passage of the replication fork [90]. Accordingly, deletion of the nuclease *EXO1*, involved in resecting telomeres 5′-3′, fully suppresses the phenotype observed [42,53]. It appears that high levels of TERRA interact with the Ku70/80 complex, resulting in an impairment of the Ku complex function in protecting chromosome ends from excessive degradation by the exonuclease Exo1. These observations, made in a context where TERRA is set experimentally high, underscore that TERRA interacts with telomeric proteins and can modulate nuclease processing activity also without forming R-loops configuration (Figure 3). Accordingly, in telomerase-positive cells, *EXO1* deletion, but not an overexpression of RNase H1, rescues the short telomere phenotype observed in a *thp2Δ* mutant [57]. In addition, R-loops have also been shown to cause sudden telomere loss events promoting telomeres shortening only when homology-directed repair (HDR) is impaired and RNase H mutants display higher levels of single-stranded telomeric DNA, suggesting a defect in replication-associated telomere processing [53,56]. 

In summary, TERRA and TERRA R-loops interfere with the replication machinery and disturb the passage of the replication fork and telomere processing, likely through several molecular mechanisms. Therefore, it is very possible that a large part of the replication stress detected at telomeres is mediated by TERRA transcription, TERRA abundance or TERRA R-loops.

## 8. TERRA and R-Loop Functions during Senescence and Post-Senescence Survival

Replicative senescence is obtained in budding yeast after the experimental inactivation of telomerase, which is otherwise constitutively expressed and active. The cell proliferation of cultures of telomerase-negative cells is first almost indistinguishable from telomerase-positive ones, but consecutive daily dilutions and re-inoculations of these cultures display a progressive decline in proliferation capability. When cultures reach about 60–80 population doublings, corresponding to 4–5 days, telomerase-negative cells start displaying a severe growth defect. Yet, if allowed to grow with fresh media for one or two more days, cultures resume normal growth. Cells in these late cultures are called post- senescent survivors. In these cells, telomere length is maintained by HDR mechanisms, also called ALT for alternative lengthening of telomeres [91,92,93,94,95]. In budding yeast, it is a rare event estimated at a frequency of 1:10,000 [92], and two types of survivors can emerge from senescent cultures: type I and type II. Type I survivors gain multiple tandem Y′ elements (even on X-only telomeres) by replication-dependent recombination and have short TG_1-3_ terminal repeats, whereas type II survivors display very heterogeneous, and sometimes extremely elevated amounts of, TG_1–3_ repeats at their chromosome ends. While the extension of these TG_1–3_ telomeric repeats by rolling circle replication of telomeric circles was detected in survivor cells, it recently became obvious that the vast majority of telomere extensions in these cells are due to break-induced replication (BIR) and that type I and type II stem from a unified BIR mechanism [92,96,97]. Rad52, a central recombination factor in budding yeast, plays an essential role in the establishment and maintenance of ALT. Most cancers re-express telomerase to elongate telomeres and maintain cell proliferation, but a substantial number of tumors are telomerase-negative. In these, telomeres are also maintained by ALT through BIR [98]. ALT mechanisms are also responsible for relapses of cancers treated with anti-telomerase drugs [99]. Therefore, understanding senescence mechanisms, and more recently, the trigger of ALT, has been the focus of much attention. As a new player in telomere biology, TERRA was found to be at the heart of ALT mechanisms in both yeast and mammalian cells.

To start with, in *S. cerevisiae*, as mentioned above, all the studies so far agree that TERRA levels and R-loops, stemming from Y′ and from X-only telomeres, increase with the shortening of telomeres in a telomerase mutant’s background [41,47,54,56,58,100,101]. Then, similarly to human cells, TERRA and telomeric hybrids were found to even be present in post-senescent survivors of telomerase-negative cultures [102,103]. Because R-loops are expected to interfere with replication and thus with telomere shortening and stability, the question is how TERRA and/or R-loops impact senescence and ALT.

In order to study how TERRA affects senescence and survivor emergence, experimental conditions affecting TERRA and telomeric R-loop levels were combined with telomerase inactivation. The first trials were the forced induction of transcription at a single telomere, which resulted in a rapid loss of viability [42,53,90]. As mentioned, this induction of TERRA transcription caused an Exo1-dependent telomere shortening in cis, and the deletion of *EXO1* fully suppressed the acceleration of senescence. Consistently, reverting the shortening of the single telomere suppressed the phenotype. This would be in accordance with a single short telomere, more so unprotected, being sufficient to trigger replicative senescence, so that cells carrying at least a very short telomere display accelerated senescence [10,12].

An independent study aiming at studying the impact of TERRA levels on senescence showed that a 262 nt C_1–3_A antisense RNA molecule is able to interact with TERRA in vivo and delay senescence [100]. This delay is independent of *EXO1* expression or the presence of HDR factors, since anti-TERRA RNA delays senescence in *exo1* or *rad52* contexts. However, this delay of senescence is epistatic with the deletion of *DOT1*, encoding an epigenetic regulator that methylates the nucleosomal histone H3Lys79. The authors concluded that Dot1 interacts with TERRA to promote senescence through its N-terminal domain, which plays a role in telomere release from nuclear periphery. Thus, TERRA could constitute a signal from the short telomeres to trigger cellular senescence through its cooperation with Dot1 [100].

To increase the levels of R-loops, mutations in the THO complex were also introduced. While *HPR1* and *THO2* deletion mutants lead to slightly longer telomeres in the presence of telomerase, when combined with telomerase inactivation, the strains displayed an accelerated senescence and a faster selection of post-senescence survivors [56,65]. This was, in principle, in accordance with the idea that R-loops would be deleterious to telomere replication and would cause telomere losses, as detected in a *THP2* deletion mutant in a telomerase-positive background [57]. Accordingly, RNAse H1 overexpression partially suppressed the accelerated senescence of the *tho2* and *hpr1* mutants [65]. However, this interpretation was questioned by the rapid appearance of post-senescent survivors. Importantly, overexpression of RNAse H2 or an RNA complementary to TERRA with the objective of decreasing the levels of TERRA R-loops impaired the appearance of post-senescence survivors, suggesting an involvement of telomeric R-loops in starting ALT [65].

In another increased R-loop context, achieved via the deletion of both RNAses H, senescence and consecutive survivor emergence were found to be delayed [56]. Conversely, the over-expression of RNAse H1 in telomerase-negative cells reduced RNA:DNA hybrids and concomitantly accelerated senescence [47,56]. These results pointed to a model whereby R-loops, as stabilized in the absence of RNAses H, prevent senescence. Furthermore, while Rad51 accumulates at the shortest telomeres in cells, together with the R-loops, the overexpression of RNase H1 abolishes both the R-loops and the Rad51 enrichment, suggesting a link between telomeric R-loops and HDR at the shortest telomeres [47]. Accordingly, the delay of senescence found in RNAse H deletion is suppressed by the deletion of *RAD52* [56]. Additionally, in agreement with these results, Npl3, an RNA binding protein involved in many RNA processes, was shown to bind TERRA and to stabilize R-loops at short telomeres, which in turn promoted HDR to prevent premature replicative senescence [101]. This is attested by the fact that the deletion of *NPL3* accelerated senescence in an epistatic manner with *RAD52* deletion, and the phenotype was suppressed when RNaseH2 was deleted. Therefore, modulating TERRA and telomeric R-loops by different means would lead to different molecular structures involving TERRA with apparent distinct consequences. Notwithstanding, RNAse H2 has been shown to be specifically recruited at telomeres, and specifically absent from short telomeres (Figure 4) [47]. Therefore, RNAse H2 depletion may better reflect the actual molecular steps at telomeres when shortening, triggering senescence and subsequently enabling ALT, probably through the stimulation of HDR, and more specifically, BIR at telomeres (Figure 4). In accordance with this model, another independent study suggests that TERRA could play a role in the recombination and amplification of Y′ elements in Type I survivors. Indeed, *paf1Δ* and *ctr9Δ* mutants, where TERRA levels increase, stimulate the induction of subtelomere Y′-element amplification in a Sir4 deficient background [104].

The next question is which mechanism can lead to such R-loop-dependent re-elongation of telomeres, in a context where R-loops are at the same time a trigger for replication stress? Rad53 is phosphorylated before the emergence of survivors, and this phosphorylation is delayed when RNase H1 is overexpressed, showing that the DNA damage checkpoint can be activated by RNA:DNA hybrids [65]. Therefore, similar to what was established in mammalian cells, TERRA transcription and R-loops may cause replication fork pauses when traversing telomeric repeats, generating appropriate substrates for BIR. In this context, the fact that RNAse H mutants display high levels of ssDNA at telomeres in an *est2 rad52* background may attest for the accumulation of such a substrate when R-loops accumulate [53]. The displaced 3′-ssDNA overhang may be extended by R-loops and coated by Rad51, resulting in a strand invasion in a replication-independent manner. This notion was further tested in mammalian cells, where TERRA R-loops were recently shown to promote ALT through an R- to D-loop switch, enabling the invasion of another telomere for BIR (Figure 4) [105].

In another vein, using cells displaying dysfunctional telomeres caused by the inactivation of the telomeric Cdc13 protein, and prone to Exo1-mediated degradation, Brian Luke’s laboratory found that the stabilization of RNA:DNA hybrids, through the loss of RNase H activity, prevents Exo1-depended telomere-end resection at short dysfunctional telomeres [106]. Thus, TERRA hybrids at telomeres could also serve to limit the extent of 5′-3′ degradations which are to increase as telomeres shorten and cells enter replicative senescence [107].

Taken together, telomeric R-loops trigger molecular mechanisms that endanger telomere semi-conservative DNA replication, enable telomere re-elongation and prevent degradation at the same time. ALT cells survive in the long term thanks to the fine-tuning of this duality [108]. In accordance with this notion, established yeast post-senescence survivors displayed growth defects. This was evidenced by careful analyses of individual sub-clones of established survivors, instead of mixed populations of cells [69]. It appeared that individual clones often experience phases of normal growth followed by severe proliferation defects. In parallel, the authors also follow the dynamics of single telomeres with unique lengths from the exact same cultures. They observe that individual telomeres undergo progressive telomere shortening up to a critical short length before being subjected to re-elongations of various lengths all of a sudden. They also correlate the extreme shortening of a given telomere with the accumulation of TERRA R-loops stemming from the short telomere in the exact same cultures. They show that the overexpression of RNAse H1 causes a more pronounced loss of viability, in accordance with the model where R-loops help in sustaining ALT and proliferation. RNAse H2 inactivation caused an inverse phenotype [69]. These results suggest that in ALT cells, the R-loop-related mechanism that triggers BIR at the shortest telomeres could well be the exact same than the one taking place in senescent cells when establishing ALT, meaning a very infrequent and perhaps deleterious event in most cases (Figure 4). These analyses also best illustrate the role of strong competition and powerful selection, masking the actual high mortality of ALT cells and unveiling clues for their fight in cancer patients.

## 9. Extratelomeric Functions of TERRA

Like other LncRNAs, TERRA seems to also play a central role in the regulation of other aspects of cell biology. In fact, the expression of many LncRNAs can change in response to stress adaptation [109]. More specifically, in mammalian cells, TERRA has been found in the extracellular inflammatory exosomes outside of the nucleus, playing a role in stimulating immune signaling, and was proposed as a marker revealing telomeric dysfunction during diseases [110,111]. Recent data also show that TERRA is induced in response to oxidative stress [112]. In budding yeast, live-cell imaging and RNA-FISH similarly show that TERRA undergoes transient induction and relocalization to the cytoplasm when yeast cells undergo diauxic shift [113]. These observations were not due to telomere shortening nor an increase in TERRA stability. These data thus suggest that a link between TERRA and an increase in oxidative stress also exist in budding yeast. While the biological significance of these observations remains to be established, these findings illustrate the complexity of TERRA biology and pave the way for new lines of research.

## 10. Conclusions

In budding yeast and other organisms, the impact of TERRA and TERRA-derived higher-order structures on telomere biology has progressed immensely since its discovery 15 years ago. Yet, much is left to understand regarding the regulation of TERRA expression, the formation of R-loops and perhaps other structures, and their biological consequences. In this review of TERRA, we focused on the unicellular simple model organism *Saccharomyces cerevisiae*, and yet, we face the complexity of TERRA regulation in time and space and the tremendous heterogeneity underlying its biology. The better understanding of TERRA is challenging, but the development of new tools to specifically modulate the expression and the structure of TERRA at individual telomeres, as well as to bypass the cell-to-cell variability in senescence phenotypes coupled to single-clone/single-cell analysis, will be indispensable to crack the heterogeneity and the apparent stochasticity of TERRA behavior. These are approaches realistic to apply in the budding yeast model.

## Figures and Tables

**Figure 1 genes-14-00618-f001:**
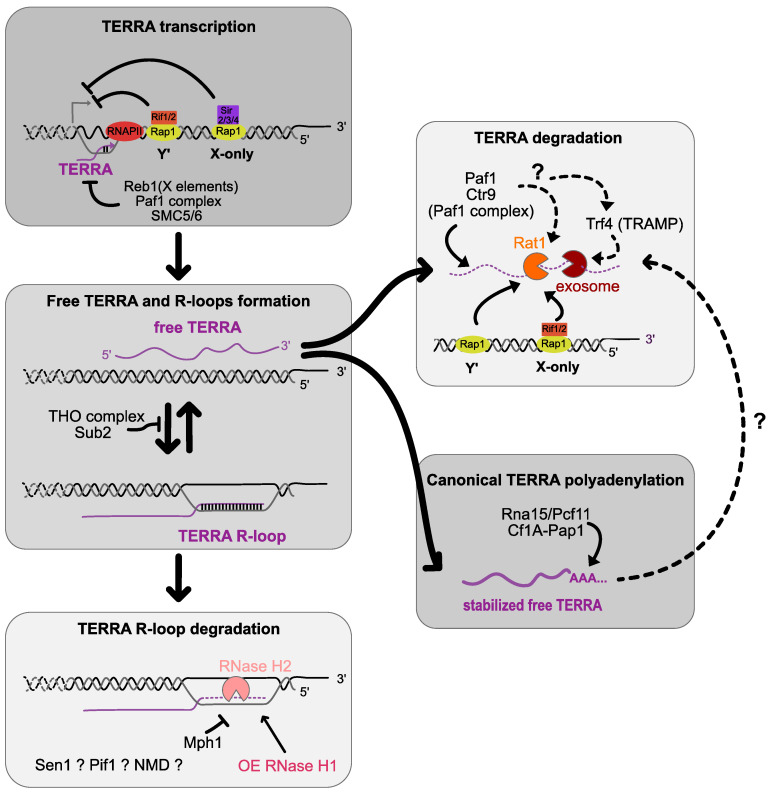
TERRA transcription and regulation in budding yeast. RNA polymerase II (RNAPII) transcribes TERRA from the 5′ C rich telomeric strand starting in subtelomeres towards telomeric ends from both X-only and Y′- telomeres. Rap1 represses TERRA transcription by recruiting Sir2/3/4 on X-only telomeres and Rif1/2 on Y′ telomeres. Reb1 binds on its binding sites residing in the X elements repressing TERRA probably by road-blocking transcription. TERRA transcription is also regulated by the Paf1 complex and SMC5/6 complex. Free TERRA is mainly degraded by the exonuclease Rat1 in manner dependent on Rap1. Rif1/2 is recruited by Rap1 and participates to the degradation of TERRA partially by recruiting Rat1 on X-only telomeres. The canonical poly(A) polymerase Pap1 stabilizes TERRA by polyadenylating its 3′ end. TRF4 is also suspected to polyadenylate TERRA which may play a role in its degradation by targeting it to the exosome. The Paf1 complex may reduce TERRA levels by acting on Rat1 or/and Trf4. TERRA is also found under an RNA:DNA configuration forming R-loops with telomeric ends. The THO complex plays a role in TERRA processing and reduces TERRA RNA:DNA hybrids. RNase H2 is found at telomeric level and degrades TERRA RNA:DNA hybrids participating in their cell cycle regulations. RNaseH1 is not detected at telomeres in telomerase positive cells but its overexpression reduces TERRA R-loops. The helicase Mph1 plays a role in limiting telomeric RNA:DNA R-loops. The involvement and local regulation of these R-loops by Sen1, Pif1 helicases and the NMD pathway has yet to be tested.

**Figure 2 genes-14-00618-f002:**
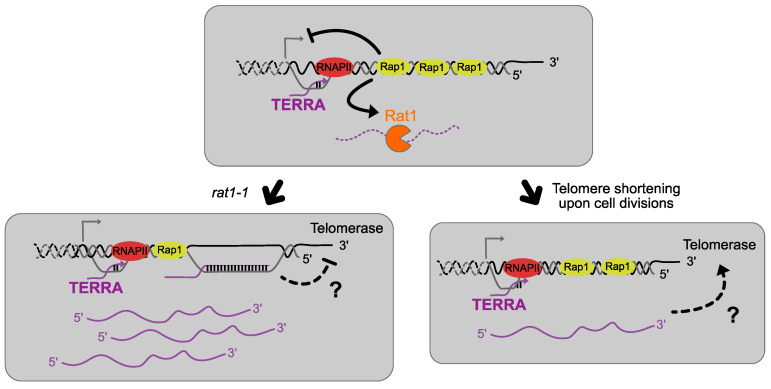
TERRA-telomerase possible interactions. In the absence of Rat1, the main exonuclease responsible for TERRA degradation, TERRA levels increase and are believed to form R-loops with telomeric ends suspected to impair the action of telomerase in elongating short telomeres. On the contrary, when telomeres shorten in a wild-type upon cell divisions, TERRA levels increase because of its *cis* regulation inhibition but TERRA is still regulated throughout the cell cycle. TERRA would then act as a scaffold molecule by interacting with telomerase to promote the elongation of the shorter telomeres during late S phase.

**Figure 3 genes-14-00618-f003:**
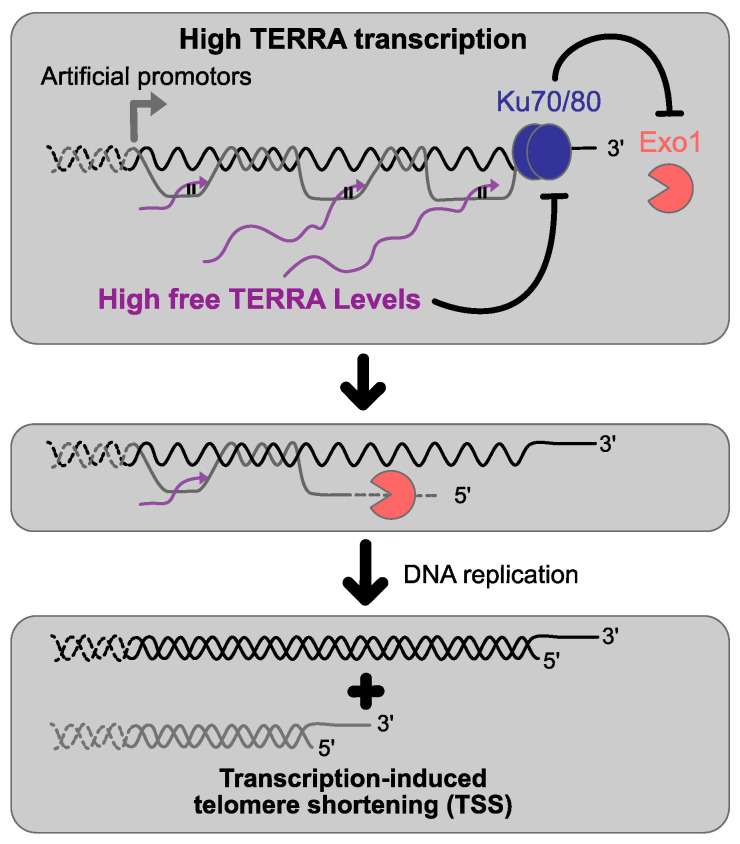
Effects of high TERRA transcription on telomere length homeostasis. The induction of TERRA transcription by artificial promoters results in high TERRA levels which promote Exo1-dependent 5′-3′ resection at telomeres by impairing the Ku complex functions resulting in transcription-induced telomere shortening in *cis*.

**Figure 4 genes-14-00618-f004:**
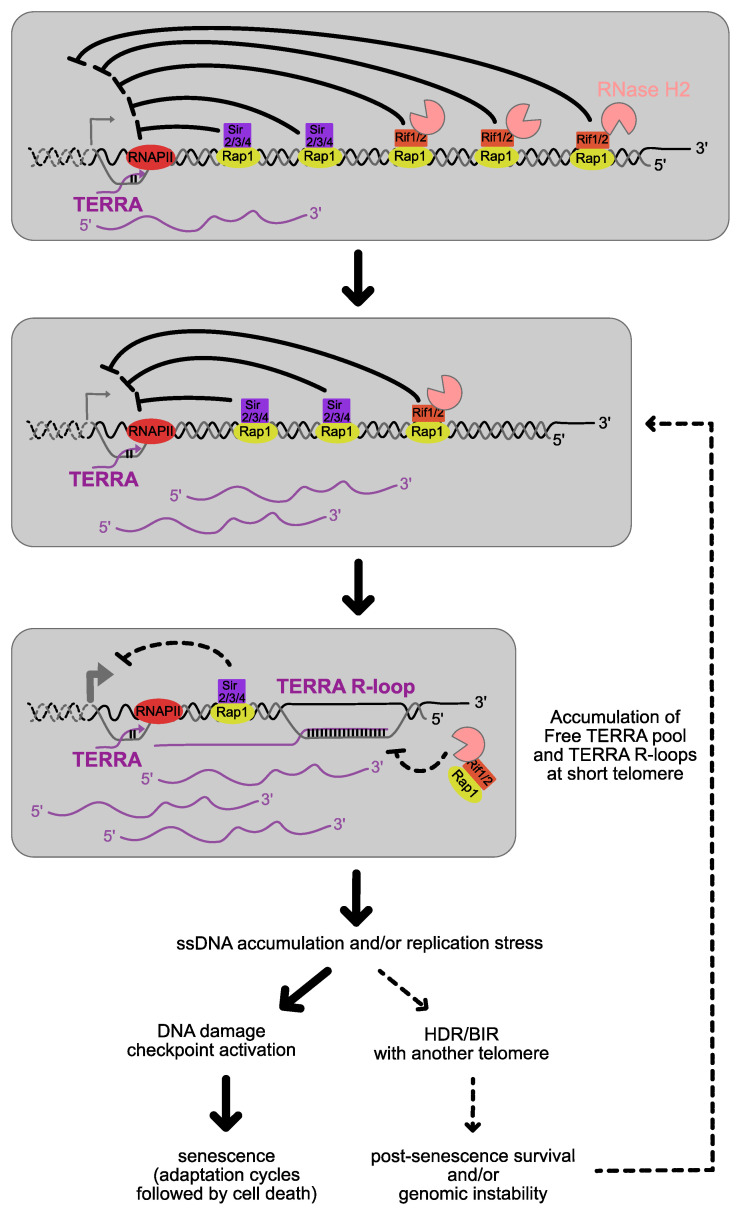
In the absence of telomerase, telomeres shorten gradually at each cell division. Free TERRA levels and TERRA R-loops increase at short telomeres due to the loss of the regulation in *cis* described in Figure 1. This generates structures that are prone to replication stress and/or recognized as DNA damage. A possible scenario is that at this point most cells undergo senescence, but in a tiny fraction the short telomeres may undergo a successful re-elongation via HDR with another longer telomere. Because these cells outgrow very fast the others, promoting or perturbing this process by introducing mutations, is expected to affect the rate of senescence. This process restarts again for each telomere reaching a critical short length.

## Data Availability

Not applicable.

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
