# Peer review of "TERRA and Telomere Maintenance in the Yeast Saccharomyces cerevisiae"

_genes, 2023, doi:10.3390/genes14030618_

Round 1
Reviewer 1 Report
The review is very nicely written and easy to follow, except from the abstract which is written in a form of brief points and needs to be rewritten. I understand the review is focused on TERRA function in budding yeast, however important findings in TERRA biology were (in addition to yeast and mammals) shown in birds (Solovei et al. Chromosome Res 1994), plants (e.g. Vrbsky et al., Plos Genet 2010, Majerova et al. Plant Mol Biol 2011, Majerova et al., Front Plant Sci 2014), or zebrafish (e.g. Park et al. Plos One 2019). These should not be ignored in the introduction and it would be also interesting if the authors point out the differences in TERRA transcription and TERRA related processes in S.cerevisiae when compared to fission yeast and higher organisms. It would be also nice to add keywords.
Author Response
"Please see the attachment."

Reviewer 2 Report
The article by Zenoun et al., entitled “TERRA and telomere maintenance in the yeast Saccharomyces cerevisiae”, submitted for publication in Genes, is a review on TERRA, non coding telomeric RNAs that play important roles in the regulation of telomere length and telomerase activity in all eukaryotic organisms and for which the yeast S. cerevisiae is one of the good model systems.
This review is extremely well documented and will undoubtfully be very useful to many readers. This is particularly remarkable as TERRA biology is extremely complex and sophisticated due to its many types of regulation and some of its yet to be iuncovered functions. Of course, these topics are of extreme importance due to their implications in telomere and telomerase functions that are crucial in cancer as well as in many telomere-related syndromes, such as aging for instance.
More specific comments:
* The paragraph entitled “TERRA and R-loop functions during senescence and post-senescence survival” (starting line 312) is very long and very difficult to read, probably due to the difficulties in interpreting the data from many papers, which are sometimes contradictory.
* Potentially, an additional Figure (Figure 4) reproduced with modification from the work of ref. 40 (inserted in text at line around 450), might represent the best example known from the literature of fine regulation of transition from telomerase-regulated telomere length to post-senescence survival regulated by TERRA at short telomeres. By doing that, this long paragraph would be easier to “digest” for the readers.
Author Response
"Please see the attachment."
